# A Sparse-Array Design Method Using Q Uniform Linear Arrays for Direction-of-Arrival Estimation

**DOI:** 10.3390/s23229116

**Published:** 2023-11-11

**Authors:** Jin Zhang, Haiyun Xu, Bin Ba, Fengtong Mei

**Affiliations:** School of Information Systems Engineering, PLA Strategic Support Force Information Engineering University, Zhengzhou 450001, China60111mei@sina.com (F.M.)

**Keywords:** sparse array, uniform linear array, cross-coarray consecutive-connected criterion, direction-of-arrival estimation

## Abstract

Nowadays, sparse arrays have been a hotspot for research in the direction of arrival (DOA). In order to achieve a big value for degrees of freedom (DOFs) using spatial smoothing methods, researchers try to use multiple uniform linear arrays (ULAs) to construct sparse arrays. But, with the number of subarrays increasing, the complexity also increases. Hence, in this paper, a design method, named as the cross-coarray consecutive-connected (4C) criterion, and the sparse array using Q ULAs (SA-UQ) are proposed. We first analyze the virtual sensor distribution of SA-U2 and extend the conclusions to SA-UQ, which is the 4C criterion. Then, we give an algorithm to solve the displacement between subarrays under the given Q ULAs. At last, we consider a special case, SA-U3. Through the analysis of DOFs, SA-UQ can find underdetermined signals. Moreover, SA-U3 can obtain DOFs close to other sparse arrays using three ULAs. The simulation experiments prove the performance of SA-UQ.

## 1. Introduction

Direction of arrival (DOA) is one of the main research hotspots in array signal processing, which is widely used in radar, sonar, and radio astronomy applications [1,2,3]. Traditional DOA estimation is applied in almost-uniform arrays, and then subspace methods are applied to obtain super-resolution results. The subspace methods, like the multiple signal classifications (MUSIC) algorithm [4], root-MUSIC algorithm [5], and estimation of signal parameters via rotational invariance techniques (ESPRIT) [6], are generally suitable for uniform arrays. But, due to the limitation of the inter-element spacing, which is no more than the half-wavelength of impinging signals, the accuracy is not high enough under the number of definite sensors.

In recent years, many sparse array structures, whose inter-element spacing can be set more than the half-wavelength of impinging signals, have been proposed. Typical sparse arrays include the minimum redundancy array (MRA) [7,8], the minimum hole array (MHA) [9,10], the nested linear array (NLA) [11,12,13], and the coprime linear array (CLA) [14,15,16]. Correspondingly, the spatial smoothing MUSIC (SS-MUSIC) and direct augmentation approach (DAA) [17] were introduced for DOA estimation. Those algorithms use the covariance matrix and construct a virtual array, which much improved the degrees of freedom (DOFs) [18], but they can only be applied to consecutive virtual sensors. Moreover, ref. [19] proved that they have the same performance. Both MRA and MHA can give the biggest aperture for a continuous virtual array. But, it was difficult to give an analytical expression about sensor location, which increased the complexity of the array design. The NLA and CLA were both composed of two uniform linear arrays (ULAs) that had the expression of the array configuration. The NLA was a hole-free array, but the CLA suffered from discontinuous virtual sensors.

In order to increase the number of consecutive virtual sensors in the CLA, a coprime array with compressed inter-element spacing (CACIS) [20] and a coprime array with displaced subarrays (CADiS) [21] were proposed. They revealed that compressing the inter-element spacing of one subarray and setting the displacement of two subarrays can improve the DOFs. Next, the shifted coprime array (SCA) [22], the coprime array with multi-period subarrays (CAMpS) [23], and the novel sparse array with two ULAs (NSA-U2) [24] were designed to find the optimal solutions for the maximum DOFs under different inter-element spacing and displacement. In conclusion, Nested CADiS [21] can have the maximum number of consecutive virtual sensors among the sparse arrays using two ULAs.

In addition to compressing and shifting processing, based on CADiS, some thoughts for filling the holes in the virtual array were proposed. The coprime array with a filled difference coarray (CAFDC), the coprime array with a two-hole difference coarray (CATHDC) in [25], and the coprime array with prime integer 3 (CAP-3) in [26] were proposed. By setting additional sensors or subarrays, the holes in CADiS can be filled. Some considered the NLA by dividing the dense subarray into several subarrays, and then gave the configurations, named the super-nested array (SNA) [27,28], the augmented nested array (ANA) [29], and the maximum inter-element spacing constraint (MISC) [30]. All these arrays had bigger DOFs than Nested CADiS. Hence, using multiple ULAs to design a sparse array can improve the DOFs compared with those only using two ULAs. But, in [29], the paper remarked that it was much more difficult to solve the sensor location when the number of subarrays was bigger than three.

Some researchers have tried to use multiple sparse arrays to construct a new sparse array. Nested MRA (NMRA) [31,32] presented the model using multiple MRAs, while the generalized nested subarray (GNSA) [33] and displaced multistage cascade subarrays (MSC-DiSA) [34] showed the models using any kind of sparse arrays. But, due to the redundancy of the same subarrays, the DOFs of this type of sparse array are lower than ANA and MISC.

Based on the analysis of the existing sparse array configurations, in this paper, we propose a sparse array design method, named the cross-coarray consecutive-connected (4C) criterion, which uses multiple ULAs. We first consider the consecutive virtual sensor distribution for any two ULAs. Then, we describe the 4C criterion and add processing for solving the displacement between subarrays under Q given ULAs, defined as a sparse array using *Q* ULAs (SA-UQ). At last, we propose a special case, SA-U3. The main contributions of this paper are as follows:The property of consecutive virtual sensors in SA-U2 is considered, which gives the conclusions corresponding to any values of sensor number and the inter-element spacing of the two subarrays.The 4C criterion is described, and one algorithm for solving displacement between subarrays under Q given ULAs is presented. The 4C criterion is less complicated than the ANA and more flexible than the GNSA. SA-UQ can estimate underdetermined signals.Based on the 4C criterion, SA-U3 is presented, which can obtain a higher DOFs than that using two ULAs and achieves DOFs close to the GNSA. Moreover, through the analysis of the design method, the order of cross-co-subarrays can influence the DOFs.

The remainder of this paper is arranged as follows. Section 2 introduces the signal model and the DOA estimation algorithm. Section 3 describes the criterion. The performance analysis and simulation experiments are presented in Section 4. Section 5 presents the conclusions of this paper.

## 2. Preliminary

### 2.1. Signal Model

Assume that a sparse array has *T* sensors located $D=\{0,d_{1},d_{2},\cdots ,d_{T-1}\}$, where the unit inter-element spacing is λ/2, and λ is the wavelength of the signals. And, there are *K* far-field narrow-band uncorrelated signals from $\theta_{k}\left(k=1,\cdots ,K\right)$ with power $\sigma_{k}^{2}$ impinging on the array. Thus, the received data are given by
(1)X=AS(t)+N,
where the manifold matrix is
(2)$$\begin{array}{cc} A & =[a\left(\theta_{1}\right),\cdots ,a\left(\theta_{K}\right)],\\ a(\theta_{k})= & {\left[1,e^{j\pi d_{1}\sin\left(\theta_{k}\right)},\cdots ,e^{j\pi d_{T-1}\sin\left(\theta_{k}\right)}\right]}^{T}, \end{array}$$
S(t) is the signals, where t=1,⋯,J and *J* is the snapshots, and N is usually Gaussian random variables with zero means and variance $\sigma_{n}^{2}$. Thus, the covariance matrix is given by
(3)$$R_{X}=XX^{H}/J=AR_{S}A^{H}+\sigma_{n}^{2}I_{T},$$
where $R_{S}=\mathrm{diag}\left[\sigma_{1}^{2},\cdots ,\sigma_{K}^{2}\right]$. The classical DOA estimation, like SS-MUSIC, indicates that the covariance matrix $R_{X}$ can be vectorized to the vector as
(4)$$Z=\mathrm{vec}\left(R_{X}\right)=Bp+\sigma_{n}^{2}e,$$
where $B=A^{*}\circ A$, ∘ is the Khatri–Rao product, e=vec(I), and p is the diagonal elements of $R_{S}$. Hence, the vector Z can be seen as the received data of a virtual linear array, whose location is defined as the difference coarray, given by
(5)$$D_{v}=\left\{d_{m}-d_{n}|d_{m},d_{n}\in D\right\}.$$

And, we define the weight function, which shows the distribution of values in $D_{v}$.

**Definition** **1.**
*The weight function of $D_{v}$ is defined as the number of coarray lag index ℓ, where <·> is cardinality of set and*

(6)
$$w\left(\ell \right)=<\left\{\left(d_{m},d_{n}\right)|d_{m}-d_{n}=\ell \right\}>\;,\;d_{m},d_{n}\in D.$$



### 2.2. SS-MUSIC

We can apply SS-MUSIC [17,19] to the virtual sensors and solve the DOA estimation. We rearrange and select the elements in Z and then have
(7)$$Z^{\prime }=FZ,$$
where F is the selection matrix with size $\left(2S_{v}+1\right)\times M^{2}$, and $2S_{v}+1$ is the maximum number of the consecutive lags. The selection matrix is given by
(8)$${\left[F\right]}_{m,a+(b-1)M}=\left\{\begin{array}{cc} 1/w(m-S_{v}-1) & m-S_{v}-1=d_{a}-d_{b}\\ 0 & \mathrm{otherwise} \end{array}\right.,$$
where $m=1,2,\cdots ,2S_{v}+1$, a=1,2,⋯,T, and b=1,2,⋯,T. Thus, $Z^{\prime }$ can be rewritten as
(9)$$Z^{\prime }=B^{\prime }p+\sigma_{n}^{2}e^{\prime }$$
where
(10)$$\begin{array}{cc} B^{\prime } & =[b^{\prime }\left(\theta_{1}\right),\cdots ,b^{\prime }\left(\theta_{K}\right)],\\ b^{\prime }\left(\theta_{k}\right)= & {\left[e^{-j\pi S_{v}\sin\left(\theta_{k}\right)},\cdots ,1,\cdots ,e^{j\pi S_{v}\sin\left(\theta_{k}\right)}\right]}^{T}, \end{array}$$
and $e^{\prime }$ is an all-zero vector except 1 located at $S_{v}+1$. Then, we divide $Z^{\prime }$ into $S_{v}+1$ parts, and each part owning $S_{v}+1$ elements is defined as ${\left[Z^{\prime }\right]}_{i:i+S_{v}}$, which means this part has the elements in $Z^{\prime }$ from index *i* to index $i+S_{v}$. Therefore, the spatial smoothing covariance matrix is
(11)$$R_{ss}=\frac{1}{S_{v}+1}\sum_{i=1}^{S_{v}+1}{\left[Z^{\prime }\right]}_{i:i+S_{v}}{\left[Z^{\prime }\right]}_{i:i+S_{v}}^{H}.$$

At last, we apply the MUSIC algorithm to $R_{ss}$ and can solve the DOA estimation.

## 3. The Configuration of SA-UQ

In this section, we try to use Q ULAs to construct a new sparse array. We assume that the *q*th subarray has $M_{q}$ sensors with inter-element spacing ${\bar{M}}_{q}$. The displacement between the first sensor in the first subarray and the first sensor in *q*th subarray is $L_{q}$, where $L_{1}=0$. The array structure is shown in Figure 1. So, the sensor location of the *q*th subarray is $D_{q}=\left\{{\bar{M}}_{q}m+L_{q}|0\leq m\leq M_{q}-1\right\}$. Moreover, the difference coarray of the *q*th and $q^{\prime }$th subarrays is
(12)$$D_{v_{qq^{\prime }}}=\left\{{\bar{M}}_{q^{\prime }}m^{\prime }-{\bar{M}}_{q}m+L_{q^{\prime }}-L_{q}|0\leq m\leq M_{q}-1,0\leq m^{\prime }\leq M_{q^{\prime }}-1\right\}.$$
where $q\neq q^{\prime }$, $D_{v_{qq^{\prime }}}$ is the cross-coarray, and when $q=q^{\prime }$, $D_{v_{qq^{\prime }}}=D_{S_{q}}$ is the self-coarray. We define a function $C\left(•\right)$, which is denoted as a new set owning all elements belonging to the maximum length of consecutive lags in one set.

### 3.1. The Basic Analysis of SA-U2

In SA-U2, we assume ${\bar{M}}_{1}$ and ${\bar{M}}_{2}$ are coprime integers, ${\bar{M}}_{1}<{\bar{M}}_{2}$, and $L_{2}=0$. Then, the cross difference coarray is given by
(13)$$D_{v_{12}}=\left\{{\bar{M}}_{2}m^{\prime }-{\bar{M}}_{1}m|0\leq m\leq M_{1}-1,0\leq m^{\prime }\leq M_{2}-1\right\},$$
and owns the following properties.

**Proposition** **1.**
*There are some facts holding for $D_{v_{12}}$.*
*1.* 
*When $M_{2}<{\bar{M}}_{1}$ or $M_{1}<{\bar{M}}_{2}$, the maximum number of consecutive lags is not bigger than ${\bar{M}}_{1}$.*
*2.* 
*When $M_{2}\geq {\bar{M}}_{1}$ and $M_{1}\geq {\bar{M}}_{2}$, the consecutive lags are from $-{\bar{M}}_{1}M_{1}+{\bar{M}}_{2}\left({\bar{M}}_{1}-1\right)+1$ to $M_{2}{\bar{M}}_{2}-{\bar{M}}_{1}\left({\bar{M}}_{2}-1\right)-1$.*
*3.* 
*With a fixed value of T, when $M_{1}=\left\lfloor T/2\right\rfloor$, $M_{2}=\left\lceil T/2\right\rceil$, ${\bar{M}}_{1}=1$ and ${\bar{M}}_{2}=M_{1}$, the number of consecutive lags, defined as $L_{v}={\bar{M}}_{1}M_{1}+{\bar{M}}_{2}M_{2}+{\bar{M}}_{1}+{\bar{M}}_{2}-2{\bar{M}}_{1}{\bar{M}}_{2}-1$, can achieve maximum value as $M_{1}M_{2}$. The values of $M_{1},M_{2}$ can be reversed, and the conclusion is still valid.*



**Proof.** See Appendix A.    □

**Remark** **1.**
*Obviously, as for SA-U2, the proposition reveals that:*
*1.* 
*In order to have a big value of $L_{v}$ in $D_{v_{12}}$, the number of sensors of the two subarrays should be no less than ${\bar{M}}_{2}$ and ${\bar{M}}_{1}$, respectively.*
*2.* 
*In order to have a big $L_{v}$ in $D_{v_{12}}$, $M_{1}$ and $M_{2}$ should be close, and the gap between ${\bar{M}}_{1}$ and ${\bar{M}}_{2}$ should be big.*



### 3.2. The Solutions for SA-UQ

We consider that the difference coarray is made up of multiple coarrays of two subarrays. Thus, we use the conclusions in Proposition 1 to construct the SA-UQ. In order to let SA-UQ have a big number of consecutive lags in $D_{v}$, we propose a design method, named the cross-coarray consecutive-connected (4C) criterion.

**Proposition** **2.**
*The 4C criterion indicates that:*
*1.* 
*Set $M_{1}=M_{2}=\cdots =M_{Q-1}=r$, and $M_{Q}=\bar{r}=T-\left(Q-1\right)r$, where r≤⌊T/Q⌋.*
*2.* 
*Set ${\bar{M}}_{Q}\leq r$ and ${\bar{M}}_{1}=1$, where ${\bar{M}}_{q}<{\bar{M}}_{Q}\left(1\leq q\leq Q-1\right)$. Moreover, any two values should be coprime integers.*
*3.* 
*The main purpose is to connect the consecutive lags of all cross difference co-subarrays. $\min\left[C\left(D_{v_{qq^{\prime }}}\right)\right]=-{\bar{M}}_{q}M_{q}+{\bar{M}}_{q^{\prime }}\left({\bar{M}}_{q}-1\right)+1+L_{q^{\prime }}-L_{q}$. and $\max\left[C\left(D_{v_{qq^{\prime }}}\right)\right]={\bar{M}}_{q^{\prime }}M_{q^{\prime }}-{\bar{M}}_{q}\left({\bar{M}}_{q^{\prime }}-1\right)-1+L_{q^{\prime }}-L_{q}$. A flow chart gives one method to find the solutions for displacement between the subarrays in Figure 2, and the key steps are shown in Algorithm 1.*



**Algorithm 1** The processing of finding the solutions for displacement between the subarrays
**Initial:** Set $q=1,q^{\prime }=2$.If $q^{\prime }=2$, let $\min\left[C\left(D_{v_{12}}\right)\right]=\max\left[C\left(\cup_{q=1}^{Q}D_{S_{q}}\right)\right]+1$, and obtain $L_{2}$. Update that $S_{v0}=\max\left[C\left(D_{v_{12}}\right)\right]$, $q=q^{\prime }$, and then $q^{\prime }=q^{\prime }+1$.If $q^{\prime }>2$ and $q=q^{\prime }-1$, let $\min\left[C\left(D_{v_{qq^{\prime }}}\right)\right]=S_{v0}+1$, and obtain $L_{q^{\prime }}$.Update that $S_{v0}=\max\left[C\left(D_{v_{qq^{\prime }}}\right)\right]$.If q>1, set q=q−1. If $S_{v0}<\min\left[C\left(D_{v_{qq^{\prime }}}\right)\right]-1$, update that $L_{q^{\prime }-1}=L_{q^{\prime }-1}-\min\left[C\left(D_{v_{qq^{\prime }}}\right)\right]+S_{v0}+1$ and $L_{q^{\prime }}=L_{q^{\prime }}-2\min\left[C\left(D_{v_{qq^{\prime }}}\right)\right]+2S_{v0}+2$. Then, turn to step 4. If $S_{v0}\geq \min\left[C\left(D_{v_{qq^{\prime }}}\right)\right]-1$, turn to step 4.If q=1 and $q^{\prime }\neq Q$, set $q=q^{\prime }$ and $q^{\prime }=q^{\prime }+1$, and turn to step 3. If q=1 and $q^{\prime }=Q$, end the processing.


**Remark** **2.**
*As for the 4C criterion, we have some remarks:*
*1.* 
*The values of the sensor number and inter-element spacing are based on Proposition 1, which ensures any cross-difference co-subarray has a big number of consecutive lags.*
*2.* 
*$D_{S_{1}}=\left\{\ell |-r+1\leq \ell \leq r-1\right\}$. So, $\max\left[C\left(U_{q=1}^{Q}D_{S_{q}}\right)\right]\geq r-1$. At first, we set $\min\left[C\left(D_{v_{12}}\right)\right]=\max\left[C\left(\cup_{q=1}^{Q}D_{S_{q}}\right)\right]+1$.*
*3.* 
*When $q^{\prime }>2$ and $q=q^{\prime }-1$, we can solve $L_{q^{\prime }}$ through $\min\left[C\left(D_{v_{qq^{\prime }}}\right)\right]=S_{v0}+1$, where $\max\left[C\left(D_{v_{1(q^{\prime }-1)}}\right)\right]+1=\min\left[C\left(D_{v_{qq^{\prime }}}\right)\right]$.*
*4.* 
*When $q^{\prime }>2$ and $1\leq q<q^{\prime }-1$, if $S_{v0}=\max\left[C\left(D_{v_{\left(q+1\right)q^{\prime }}}\right)\right]<\min\left[C\left(D_{v_{qq^{\prime }}}\right)\right]$, two consecutive parts are not connected. Thus, we change the value $L_{q^{\prime }-1}=L_{q^{\prime }-1}-\min\left[C\left(D_{v_{qq^{\prime }}}\right)\right]+S_{v0}+1$ and $L_{q^{\prime }}=L_{q^{\prime }}-2\min\left[C\left(D_{v_{qq^{\prime }}}\right)\right]+2S_{v0}+2$ to ensure that two sets are continuous.*



For the sake of clarity, we propose an example that T=20, Q=4, $M_{q}=5$, ${\bar{M}}_{q}=\left\{1,3,4,5\right\}$. Thus, $\max\left[C\left(\cup_{q=1}^{Q}D_{S_{q}}\right)\right]=6$, and the other steps are shown in Table 1.

### 3.3. The Application in SA-U3

In this subsection, we take the consideration of SA-U3, where
(14)$$\begin{array}{cc} D_{1} & =\{m|0\leq m\leq r-1\},\\ D_{2} & =\{2m+L_{2}|0\leq m\leq r-1\},\\ D_{3} & =\{rm+L_{3}|0\leq m\leq \bar{r}\}, \end{array}$$
r=2[T/6]−1, $\bar{r}=T-2r$, and [•] is denoted as the nearest integer of a value. Under this set of inter-element spacing and the sensor number of subarrays, we use the algorithm in Figure 2 and solve $L_{2}=2r+1,L_{3}=7r$, and $S_{v}=\left(\bar{r}-1\right)r+7r$.

But, it can be improved. Based on the 4C criterion, we set $L_{2}=\left(\bar{r}+2\right)r-1$ and $L_{3}=\left(\bar{r}+4\right)r-3$. Here, $D_{2}$ and $D_{3}$ have one common sensor located at $L_{3}$. Thus, the difference coarray $D_{v}$ has the following properties:

**Proposition** **3.**
*The distribution of consecutive lags in $D_{v_{ij}}$ satisfies that:*
*1.* 
*$D_{S}=\cup_{q=1}^{3}D_{S_{q}}$ has the consecutive lags from −r+1 to r+1.*
*2.* 
*$D_{v_{12}}$ has the consecutive lags from $-r+1+L_{2}$ to $2\left(r-1\right)+L_{2}$.*
*3.* 
*$D_{v_{13}}$ has the consecutive lags from $-r+1+L_{3}$ to $\left(\bar{r}-1\right)r+L_{3}$.*
*4.* 
*$D_{v_{23}}$ has the consecutive lags from $-r+1+L_{3}-L_{2}$ to $\bar{r}r-r+1+L_{3}-L_{2}$.*
*5.* 
*$D_{v}$ has the consecutive lags from $-S_{v}$ to $S_{v}$, where $S_{v}=2\bar{r}r+4r-3$.*



**Proof.** The self-difference coarray can be denoted as
(15)$$D_{S}=\left\{\pm m|0\leq m\leq r-1\right\}\cup \left\{\pm 2m|0\leq m\leq r-1\right\}\cup \left\{\pm rm|0\leq m\leq \bar{r}-1\right\}.$$Moreover, based on Proposition 1, we can have the values of consecutive lags in $D_{v_{ij}}$.
From (Equation 15), it is obvious that the lags from −r+1 to r−1 belong to $D_{S_{1}}$, the lags ±r belong to $D_{S_{3}}$, and the lags ±(r+1) are even and belong to $D_{S_{2}}$. So, $\max\left[C\left(\cup_{q=1}^{Q}D_{S_{q}}\right)\right]=r+1$.The subarray 1 and subarray 2, respectively, have the inter-element spacing as 1 and 2, and both have *r* sensors. Thus, $\bar{M}=1,\bar{N}=2,M=N=r$. $D_{v_{12}}$ has no hole and has the lags from $-r+1+L_{2}$ to $2\left(r-1\right)+L_{2}$. Due to $L_{2}=\left(\bar{r}+2\right)r-1$, the range of lags is $\bar{r}r+r\leq \ell \leq \bar{r}r+4r-3$.The subarray 1 and subarray 3 have the inter-element spacing as 1 and *r* with *r* and $\bar{r}$ sensors. Thus, $\bar{M}=1,\bar{N}=N=r,M=\bar{r}+1$. $D_{v_{13}}$ has no hole and has the lags from $-r+1+L_{3}$ to $r\left(\bar{r}-1\right)+L_{3}$. Due to $L_{3}=\left(\bar{r}+4\right)r-3$, the range of lags is $\bar{r}r+3r-2\leq \ell \leq 2\bar{r}r+4r-3$.The subarray 2 and subarray 3 have the inter-element spacing as 2 and *r* with *r* and $\bar{r}$ sensors. Thus, $\bar{M}=2,\bar{N}=N=r,M=\bar{r}+1$. $D_{v_{23}}$ has the consecutive lags from $-r+1+L_{3}-L_{2}$ to $\bar{r}r-r+1+L_{3}-L_{2}$. Due to $L_{3}-L_{2}=2r-2$, the range of consecutive lags is $r-1\leq \ell \leq \bar{r}r+r-1$.Based on the proof of the last four properties, we can find that the the consecutive of the self-difference coarray and cross-difference coarray are connected, satisfying the 4C criterion. Hence, $S_{v}=2\bar{r}r+4r-3$.In conclusion, all properties are true. Moreover, SA-U3 has no hole in the difference coarray. □

Obviously, when r>3, $2\bar{r}r+4r-3-\left[\left(\bar{r}-1\right)r+7r\right]=\bar{r}r-3r-3>0$. Although two configurations both satisfy the 4C criterion, the improved one changes the order of the co-subarrays, which is $D_{v_{23}}\to D_{v_{12}}\to D_{v_{13}}$, while the original one is $D_{v_{12}}\to D_{v_{23}}\to D_{v_{13}}$. Hence, the order of the co-subarrays can influence the value of $S_{v}$. Moreover, when Q=3, we only need to consider the order of $D_{v_{12}}$ and $D_{v_{23}}$. But, when Q>3, it is more difficult to analyze and decide the order. Considering that SA-UQ proposed by the algorithm in Figure 2 has already obtained a big $S_{v}$, this algorithm is still effective for Q>3.

In this paper, except that SA-U3 is the improved one, another SA-UQ is proposed using the algorithm in Figure 2. The expression of the consecutive lags is provided, which is important for using SS-MUSIC. Through SS-MUSIC, we can solve the DOA estimation.

## 4. Performance Analysis and Simulation Experiments

### 4.1. The Analysis of DOF

In this part, we discuss the DOFs of sparse arrays. When we use SS-MUSIC to solve DOA estimation, the DOFs are equivalent to $S_{v}$. At first, we compare the DOFs of SA-U3 with those of MRA [7], NLA [11], Nested CADiS [21], CAP-3 [26], ANAI-1, ANAII-1 [29], and MISC [30]. We vary *T* from 9 to 18. The results are shown in Figure 3. All the sparse arrays are composed of several ULAs. The MRA has the biggest value of $S_{v}$. When *T* is small, ANAII-1, which uses five ULAs, can have the same DOFs as MRA. And, MISC using four ULAs has the second biggest value of $S_{v}$. We can find that the the proposed array SA-U3 has DOFs close to ANAI-1 and CAP-3, which are all composed of three ULAs, demonstrating the efficiency of the design method. Moreover, SA-U3 has much improved DOFs compared with Nested CADiS and the NLA.

Then, we compare the SA-UQ with the GNSA. We vary *T* from 9 to 45 with three intervals. As for GNSA, we choose the MRA+MRA design model [33], and the number of subarray sensors is $\left\lfloor \sqrt{T}\right\rfloor$, which can produce the biggest number of consecutive lags in the difference coarray. As for SA-UQ, we vary Q={3,4,5}. As for SA-U4 and SA-U5, r=⌊T/Q⌋. When Q=4, if T={21,24,27}, ${\bar{M}}_{q}=\left\{1,2,3,5\right\}$; if T={30,33,36,39}, ${\bar{M}}_{q}=\left\{1,5,6,7\right\}$; and if T={42,45}, ${\bar{M}}_{q}=\left\{1,7,8,9\right\}$. When Q=5, if T={36,39,42,45}, ${\bar{M}}_{q}=\left\{1,2,3,5,7\right\}$. The results are shown in Figure 4. Here, we can find that SA-U3 can have the $S_{v}$ close to the GNSA. But, considering that the GNSA cannot be applied to any value of *T*, the 4C criterion is a more flexible design method. Moreover, comparing SA-UQ with different *Q*, we can obtain the lower $S_{v}$ with the increasing *Q*.

### 4.2. Simulation Results

To verify the DOA estimation performance of the proposed array configuration, we simulate the performance of SA-UQ and compare it with other sparse arrays. We compare the performance using the root mean square error (RMSE) to quantify the accuracy, which is defined as
(16)$$\mathrm{RMSE}=\sqrt{\frac{1}{GK}\sum_{m=1}^{G}\sum_{g=1}^{K}{\left({\hat{\theta }}_{k,g}-\theta_{k}\right)}^{2}}$$
where *G* is the number of Monte Carlo simulations, *K* is the number of signals, and ${\hat{\theta }}_{k,g}$ is the DOA of the kth signal in the gth Monte Carlo simulation.

Simulation 1: MUSIC spectrum under the underdetermined signals.

Using SS-MUSIC, we can find underdetermined signals in sparse arrays. Thus, we conduct the simulations in SA-U3 and SA-U4 with T=20. As for SA-U4, r=5, ${\bar{M}}_{q}=\left\{1,3,4,5\right\}$. Set the SNR=0 dB and J=5000. And, we respectively assume that K=25 and K=35, where the signals are uniformly distributed from $-45^{\circ }$ to $45^{\circ }$. The normalized spatial spectrum is presented in Figure 5. The simulation results show that both SA-U3 and SA-U4 can effectively estimate all DOAs under low SNR.

Simulation 2: RMSE comparison of SA-UQ and the existing sparse arrays under different SNRs.

In this simulation, we compare the RMSE of SA-UQ with those of Nested CADiS, GNSA, CAP-3, ANAI-1, ANAII-1, and MISC. We set T=20, J=5000, and vary SNR from −5 dB to 15 dB with 5 dB intervals. Assume that K=25, where the signals are uniformly distributed from $-45^{\circ }$ to $45^{\circ }$. We conduct G=500 simulation times and the results are shown in Figure 6. With the increase in the SNR, the RMSEs decrease and decrease gradually more slowly. When considering the results of comparison of $S_{v}$, we can find that the sparse array with bigger $S_{v}$ can have the lower RMSE using SS-MUSIC. Thus, ANAII-1 and MISC have the highest accuracy. SA-U3 has an RMSE close to that of ANA-1. SA-U4 and Nested CADiS have the lowest $S_{v}$, so they have the biggest RMSEs.

Simulation 3: RMSE comparison of SA-UQ and the existing sparse arrays under different snapshots.

Based on Simulation 2, we set a fixed value for the SNR as 0dB, and vary *J* from {20,50,100,200,500,1000,2000,5000}. The results are presented in Figure 7. The simulation results show that as the number of snapshots increases, the RMSEs decrease. When J≥1000, the RMSEs become flattened. Comparing the RMSEs of different sparse arrays, we can have the same conclusions as for Simulation 2.

## 5. Conclusions

In this paper, a new sparse array design method, named as the 4C criterion, and configuration, defined as SA-UQ, which uses Q ULAs, are proposed. The paper first gives the property of consecutive lags in SA-U2 and concludes the requirement for the values of both inter-element spacing and sensor number for any two ULAs. Then, the conclusions extend to sparse arrays using Q ULAs, which is the 4C criterion, and the method for finding solutions for the displacement between subarrays in SA-UQ is presented. Lastly, the paper proposes SA-U3 and gives the analytical expression for sensor location. Through the analysis of the difference coarray, the 4C criterion is effective. Moreover, the simulation experiments prove that SA-UQ can estimate underdetermined signals, and SA-U3 can have RMSEs close to the other sparse arrays using three ULAs.

## Figures and Tables

**Figure 1 sensors-23-09116-f001:**
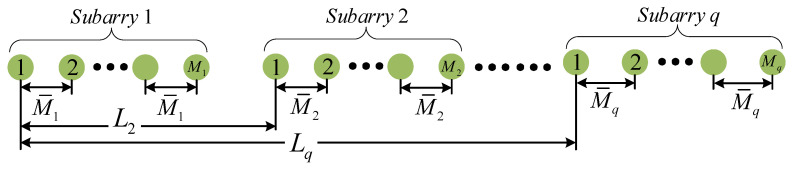
The array structure of SA-UQ.

**Figure 2 sensors-23-09116-f002:**
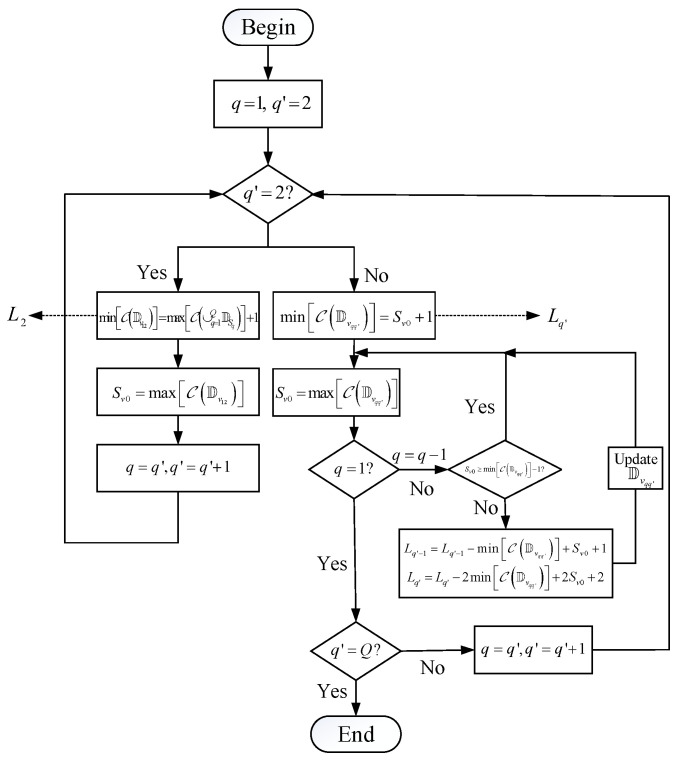
The flowchart of the processing to find solutions for SA-UQ.

**Figure 3 sensors-23-09116-f003:**
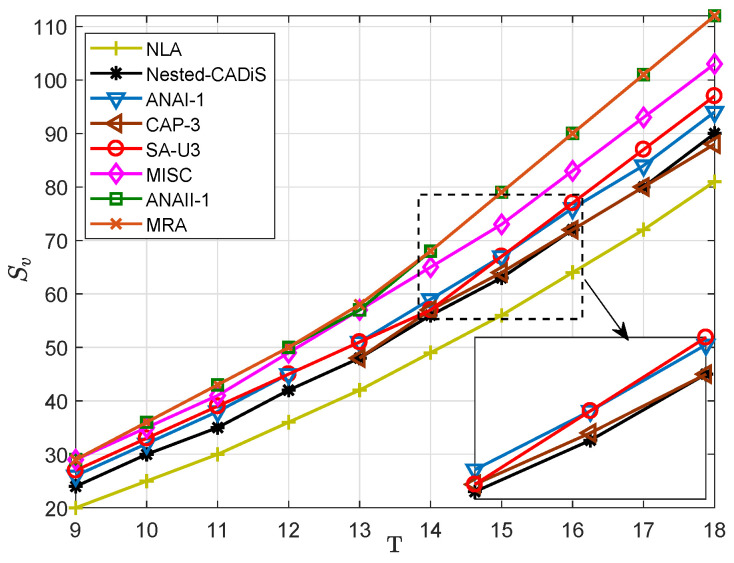
$S_{v}$ in different sparse arrays with different numbers of sensors, *T*.

**Figure 4 sensors-23-09116-f004:**
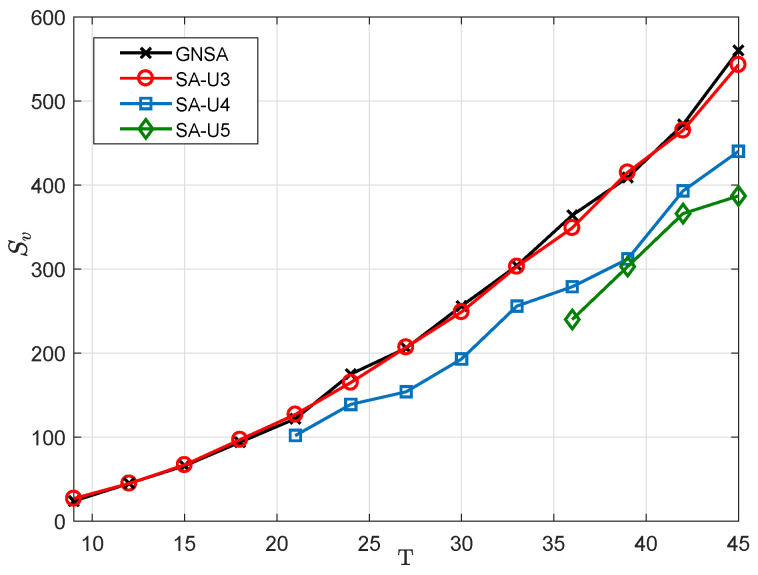
$S_{v}$ in SA-UQ and GNSA with different numbers of sensors, *T*.

**Figure 5 sensors-23-09116-f005:**
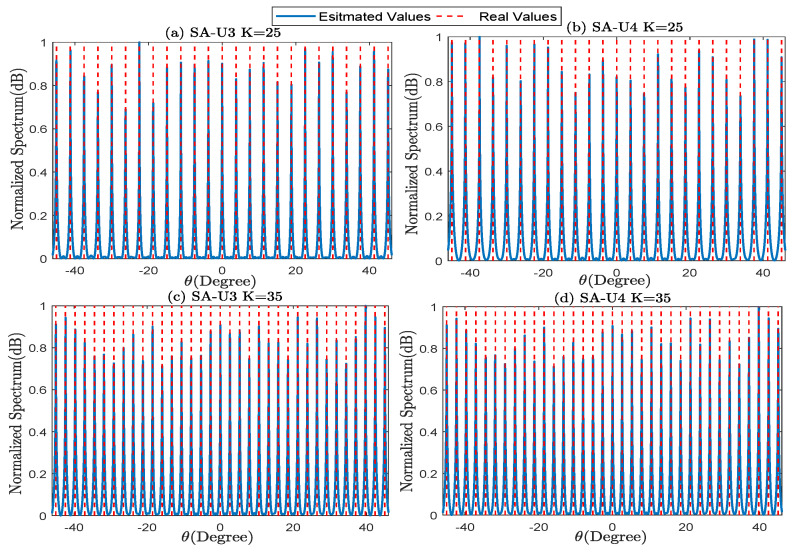
The spatial spectrum of (**a**) K=25 in SA-U3, (**b**) K=25 in SA-U4, (**c**) K=35 in SA-U3, (**d**) K=35 in SA-U4.

**Figure 6 sensors-23-09116-f006:**
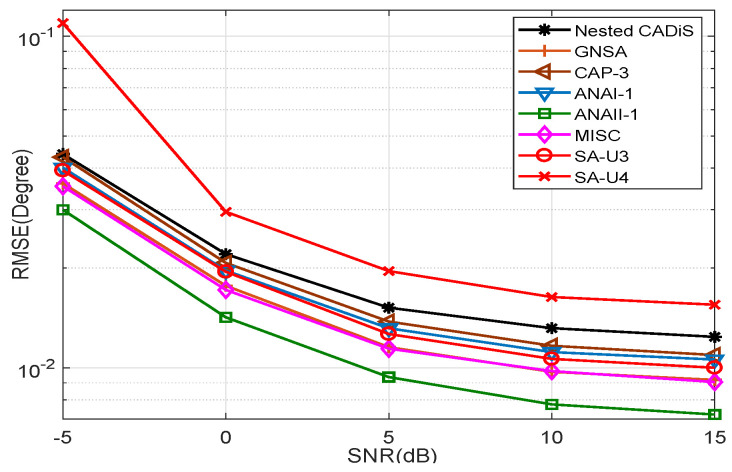
The comparison of RMSEs in different sparse arrays under different SNRs.

**Figure 7 sensors-23-09116-f007:**
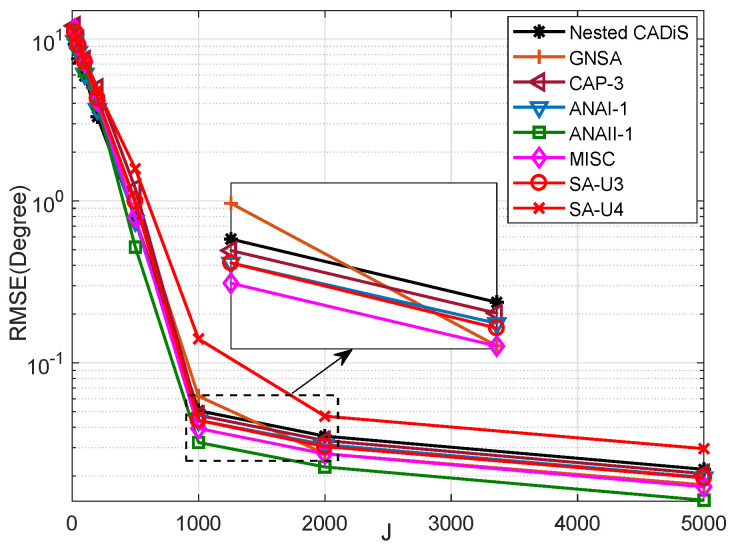
The comparison of RMSEs in different sparse arrays under different snapshots.

**Table 1 sensors-23-09116-t001:** The detailed steps to find solutions for SA−U4 with T=20.

$q^{\prime }$	$L_{q^{\prime }}$	$C\left(D_{v_{{\mathrm{qq}}^{\prime }}}\right)$
$q^{\prime }=2$	$\min\left[C\left(D_{v_{12}}\right)\right]=-4+L_{2}=7$	$C\left(D_{v_{12}}\right)=\left[7,23\right]$
	→$L_{2}=11$	
$q^{\prime }=3$	$\min\left[C\left(D_{v_{23}}\right)\right]=-6+L_{3}-L_{2}=24$	$C\left(D_{v_{23}}\right)=\left[24,40\right]$
	→$L_{3}=41,L_{3}-L_{2}=30$	
	→$L_{3}-L_{1}=41$	$C\left(D_{v_{13}}\right)=\left[37,57\right]$
$q^{\prime }=4$	$\min\left[C\left(D_{v_{43}}\right)\right]=-4+L_{4}-L_{3}=58$	$C\left(D_{v_{34}}\right)=\left[58,70\right]$
	→$L_{4}=103,L_{4}-L_{3}=62$	
	→$L_{4}-L_{2}=92$	$C\left(D_{v_{24}}\right)=\left[88,104\right]$
Update	$L_{3}=41-\left(88-70-1\right)=24$	$C\left(D_{v_{23}}\right)=\left[8,24\right]$
	$L_{4}=103-2\left(88-70-1\right)=69$	$C\left(D_{v_{13}}\right)=\left[20,40\right]$
	→$L_{4}-L_{3}=45$	$C\left(D_{v_{34}}\right)=\left[41,53\right]$
$q^{\prime }=4$	→$L_{4}-L_{2}=58$	$C\left(D_{v_{24}}\right)=\left[54,70\right]$
	→$L_{4}-L_{1}=69$	$C\left(D_{v_{14}}\right)=\left[65,89\right]$

## Data Availability

The authors claim that the data used in this article are provided by our simulations and the data used to support the findings of this study are available from the corresponding author upon request.

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
