# Peer review of "A Sparse-Array Design Method Using Q Uniform Linear Arrays for Direction-of-Arrival Estimation"

_sensors, 2023, doi:10.3390/s23229116_

Round 1
Reviewer 1 Report
Comments and Suggestions for Authors
The author discusses the sparse array design method using Q uniform linear arrays for DOA estimation. Overall the work is technically good and easy to follow, here are my minor suggestions:
1) What is the meaning of the symbol placed in line no. 137?
2) Is there any standard reference to fix criteria 4c?
3) It is very difficult to follow Figure 1. Please make it more appropiate
4) Figure 2 and 3, what is the x-axis?
5) Can you explain what is J in Figure 6?
6) Compare your work with similar type available in literature and then list the significance of your work over them.
Comments on the Quality of English LanguageMinor editing of English language required
Author Response
Dear reviewer,
Thank you for your reasonable, scientific and patient review, which makes me benefit a lot from more rigorous scientific research. With regard to your valuable Suggestions, we reply as follows:
1) I'm sorry that the symbol placed in line no.137 has brought you any confusion. This symbol is used to represent the end of the mathematical proof. This symbol appears in the LaTeX template provided by this journal when the mathematical proof is used.
2) Thank you for your question. As for the revision of the 4c criterion, it is mainly through step (e) in the Algorithm 1 on page 5. This manuscript follows the maximization of continuous virtual array sensors. Because the virtual array is judged to be discontinuous, so the numerical update of the subarray spacing is carried out.
3) I'm sorry that the Figure 1 has brought you any confusion. Figure 1 is a description of the processing flow of the proposed algorithm, and to make it more appropriate, we add a flow in the figure: updating Dvqq' value. we hope to make your reading more appropriate.
4) I'm sorry that Figure 2 and 3 have brought you any confusion. In Figure 2 and 3, the x-axis represents the total number of array sensors.
5) I'm sorry that the J in Figure 6 has brought you any confusion. In Figure 6, J represents the number of snapshots. Figure 6 shows the comparison of RMSE in different sparse array under different snapshots.
6) I'm sorry for any confusion. In this manuscript, a new sparse array design method is presented, and its performance is compared with similar type available in literature. For example, MRA [7], NLA [11], Nested-CADiS [21], CAP-3 [26], ANAI-1, ANAII-1 [29] and MISC [30]. In the section of Performance analysis and simulation experiments, the degree of freedom (DOF) and root mean square error (RMSE) of this manuscript and similar type available in literature are compared. Through the analysis of difference coarray, 4C criterion is effective. Through the analysis of DOF, SA-UQ can find underdetermined signals. Moreover, the simulation experiments prove the performance of SA-UQ, and SA-U3 can have the close RMSE as the other sparse arrays using 3 ULAs.
Thank you again for your valuable suggestions.
Yours sincerely!

Reviewer 2 Report
Comments and Suggestions for Authors
Please find the document attached.

Comments on the Quality of English LanguageAuthor Response
Dear reviewer,
Thank you for your reasonable, scientific and patient review, which makes me benefit a lot from more rigorous scientific research. With regard to your valuable Suggestions, we reply as follows:
1) Thank you very much for pointing out our oversight in the article. We have given the complete form of uniform linear arrays (ULAs) in the manuscript.
2) Thank you for your valuable comment. We are very sorry that the grammatical errors in the text. We have corrected some grammatical errors in the text. We will continue to modify the grammar of the manuscript. Thank you again for your valuable comment.
3) I'm sorry that the square box has brought you any confusion. This square box is used to represent the end of the mathematical proof. This square box appears in the LaTeX template provided by this journal when the mathematical proof title has been mentioned.
Thank you again for your valuable comments.
Yours sincerely!